# Bayesian rank likelihood-based estimation: An application to low birth weight in Ethiopia

Daniel Biftu Bekalo[1,2]*, Anthony Kibira Wanjoya[3], Samuel Musili Mwalili[3]

**1** Pan African University Institute for Basic Sciences, Technology and Innovation, Nairobi, Kenya,
**2** Haramaya University, Dire Dawa, Ethiopia, **3** Jomo Kenyatta University of Agriculture and Technology, Nairobi, Kenya

\* danibiftu@gmail.com

**Data Availability Statement:** The data used for the current study is publicly available on the DRYAD repository, and it can be accessed with the following link: https://doi.org/10.5061/dryad.3j9kd51sg.

## Abstract

### Background

Low birth weight is a significant risk factor associated with high rates of neonatal and infant mortality, particularly in developing countries. However, most studies conducted on this topic in Ethiopia have small sample sizes, often focusing on specific areas and using standard models employing maximum likelihood estimation, leading to potential bias and inaccurate coverage probability.

### Methods

This study used a novel approach, the Bayesian rank likelihood method, within a latent traits model, to estimate parameters and provide a nationwide estimate of low birth weight and its risk factors in Ethiopia. Data from the Ethiopian Demographic and Health Survey (EDHS) of 2016 were used as a data source for the study. Data stratified all regions into urban and rural areas. Among 15, 680 representative selected households, the analysis included complete cases from 10, 641 children (0-59 months). The evaluation of model performance considered metrics such as the root mean square error, the mean absolute error, and the probability coverage of the corresponding 95% confidence intervals of the estimates.

### Results

Based on the values of root mean square error, mean absolute error, and probability coverage, the estimates obtained from the proposed model outperform the classical estimates. According to the result, 40.92% of the children were born with low birth weight. The study also found that low birth weight is unevenly distributed across different regions of the country with the highest amounts of variation observed in the Afar, Somali and Southern Nations, Nationalities, and Peoples regions as represented by the latent trait parameter of the model. In contrast, the lowest low birth weight variation was recorded in the Addis Ababa, Dire Dawa, and Amhara regions. Furthermore, there were significant associations between birth weight and several factors, including the age of the mother, number of antenatal care visits, order of birth and the body mass index as indicated by the average posterior beta values of

**Funding:** The author(s) received no specific funding for this work.

**Competing interests:** The authors have declared that no competing interests exist.

**Abbreviations:** ANC, Antenatal Care; BMI, Body Mass Index; CI, Credible Interval; EA, Enumeration Areas; EDHS, Ethiopian Demographic and Health Survey; LBW, Low Birth Weight; MAE, Mean Absolute Error; MCMC, Markov Chain Monte Carlo; MSE, Mean Squared Error; PC, Probability Covarage; PSRF, Potential Scale Reduction Factor; RMSE, Root Mean Squared Error; SNNPR, Southern nations, nationalities, and people representative of Ethiopia; WHO, World Health Organization.

$(\beta_1 = -0.269, CI=-0.320, -0.220)$, $(\beta_2 = -0.235, CI=-0.268, -0.202)$, $(\beta_3 = -0.120, CI=-0.162, -0.074)$ and $(\beta_5 = -0.257, CI=-0.291, -0.225)$.

## Conclusions

The study showed that the low birth weight estimates obtained from the latent trait model outperform the classical estimates. The study also revealed that the prevalence of low birth weight varies between different regions of the country, indicating the need for targeted interventions in areas with a higher prevalence. To effectively reduce the prevalence of low birth weight and improve maternal and child health outcomes, it is important to concentrate efforts on regions with a higher burden of low birth weight. This will help implement interventions that are tailored to the unique challenges and needs of each area. Health institutions should take measures to reduce low birth weight, with a special focus on the factors identified in this study.

## Background

The World Health Organization (WHO) has defined low birth weight (LBW) as a condition in which babies weigh less than 2500 grams at birth, regardless of their gestational age [1]. Furthermore, the Centers for Disease Control have classified birth weight into several classifications such as; extremely low birth weight (ELBW), very low birth weight (VLBW), low birth weight (LBW), normal birth weight (NBW) and high birth weight (HBW) [2]. Infants weighing less than 1,000 grams, less than 1,500 grams, less than 2,500 grams, less than 4,000 grams and greater than 4,000 grams are classified as ELBW, VLBW, LBW, NBW, and HBW, respectively.

Each year, more than 20 million babies are born around the world with low birth weight. Developing countries, especially those in sub-Saharan Africa, such as Ethiopia, are disproportionately affected, with most cases occurring in this region [3]. In Africa as a whole, 22% of babies are born with low birth weight, and in sub-Saharan Africa, this figure ranges from 13% to 15%, with slight variations from region to region [4]. According to the Ethiopian Demographic and Health Survey (EDHS 2016), low birth weight affects 26% of newborns born in Ethiopia [5].

In sub-Saharan African nations, LBW is a notable indicator of prenatal morbidity and mortality. Furthermore, LBW is associated with a variety of short and long term outcomes, including the development of chronic diseases later in life [6]. Infants with extremely low, very low, or low birth weight are prone to complications such as respiratory distress, anemia, poor nutrition, infections, neurological problems, and hearing impairments [7]. Previous Ethiopian studies found significant associations between various factors and low birth weight. These factors include inadequate weight gain during pregnancy, low body mass index, short birth interval, demanding physical work during pregnancy, illness (particularly infections), birth order, and lack of adequate antenatal care (ANC) [8–11].

Several studies have been carried out in Ethiopia to assess the occurrence and factors that influence the occurrence of LBW. The occurrence of LBW in Ethiopia, as reported in different studies, varied between 6% and 29.1% [8, 12–15]. However, most of these studies used the maximum likelihood method to estimate parameters linked to factors related to LBW. According to [16], the presence of a latent variable in the ordinal response variable leads to biased

estimates and inaccurate coverage probabilities with the method based on maximum likelihood. Furthermore, most of the studies carried out in Ethiopia used classical models with limited sample sizes, focusing on specific regions or areas. Therefore, this study was carried out to introduce a novel approach, the Bayesian rank likelihood method in the latent traits model, to estimate parameters and provide an estimate of LBW and its related factors in Ethiopia.

Rank likelihood has been used in models for the statistical analysis of various problems, ranging from the analysis of paired comparisons to randomly censored survival data. Ranks, rather than the original values of the observations, if available, prove useful in statistical analysis if there are no strong prior beliefs about the distributional properties of the observations or if the distributional properties of the observations are difficult to assess [17]. This likelihood was first suggested for continuous variables by [17, 18] investigated its theoretical characteristics.

The results of this study would be crucial in formulating improved health policies aimed at preventing LBW and developing targeted strategies to address identified factors. Furthermore, it will help the country by providing informative data that can identify areas requiring intervention and contribute to the achievement of national and international goals and goals related to maternal and infant health.

## Methods

### Study area

Ethiopia is organized into four administrative levels, namely, region, zone, woreda and kebele. The primary administrative division in Ethiopia is the region, which is sometimes referred to as a kilil or regional state. There are twelve regions, namely Tigray, Afar, Amhara, Oromia, Benishangul-Gumuz, Somali, Gambela, Harari, Sidama, Central Ethiopia, South Ethiopia and South West Ethiopia, along with two independently administered cities, Addis Ababa and Dire Dawa. The determination of regions in Ethiopia is based on ethnolinguistic areas.

### Data source

The study focused on children under five years of age in Ethiopia and used the most recent standard and nationally representative data from the Ethiopian Demographic and Health Survey (EDHS) of 2016, which is the fourth nationally representative survey conducted in Ethiopia as part of the Demographic and Health Surveys project worldwide. The primary objective of the survey was to provide accurate and timely information on health and demographic outcomes at the national and regional levels [5]. Data for EDHS 2016 were taken from the DHS website (http://dhsprogram.com) after obtaining permission. Additional information on the design of the EDHS survey and LBW data is summarized in [5].

### Study design and settings

EDHS 2016 was conducted using standardized survey design and data collection procedures. EDHS 2016 used a two-stage stratified cluster sampling method to select enumeration areas (EA) proportionally based on their sizes. Subsequently, a random selection of households was made from the selected enumeration areas (EA). Data were collected through face-to-face interviews conducted with consented household members using questionnaires. Data collection for EDHS 2016 was carried out from January 18 to June 27, 2016. EDHS 2016 was structured to provide representative data on health and demographic indicators at the national, urban, rural, and regional levels. LBW measurements were taken for all children under five years of age in selected households. The study aimed to evaluate the low birth weight status of

children under five years of age by measuring their weights at birth. Among 15, 680 representative selected households in the 2016 EDHS, only complete cases of 10, 641 children (0–59 months) were used.

## Study variables

The independent variables used in this study were taken from previous research on child birth weight, as indicated by the studies [3, 19–21]. In this study, the variable of interest was the birth weight of the child, which served as the response variable. The covariates considered in this study included the age of the mother in years, the body mass index of the mother, the order of birth, the preceding birth interval in months, the number of visits to antenatal care, and the region where the child was born.

## Method of data analysis

**Ordered probit model.** A popular method of estimating the ordered response variable is the ordered probit model, which utilizes the probit link. This model, like many others for qualitative dependent variables, originated in biostatistics [22]. The main concept is that there exists an underlying continuous metric that is hidden or not directly observed. This metric is the basis for the ordinal responses observed by the analyst. The metric is divided into various categories based on specific thresholds. In ordered probit regression, the response variable $Y$ is related to a predictor vector $X$ through a regression that involves a latent variable $Z$. To further elaborate, the model can be expressed as follows:

$$\epsilon_1, \ldots, \epsilon_n \overset{\text{iid}}{\sim} N(0, 1)$$

$$Z_i = X_i^T \beta + \varepsilon_i$$

$$Y_i = g(Z_i)$$

where $\beta$ and $g$ are unknown parameters.

In the probit regression model, according to [23], it is assumed that the variance of $\varepsilon_1, \ldots, \varepsilon_n$ is one.

**Latent traits model.** In this study, we employ a Bayesian approach that incorporates the rank likelihood estimation within a latent trait model to estimate low birth weight (LBW). Our work emphasizes the integration of a Bayesian method into the framework of the latent traits model to incorporate the rank likelihood method for LBW estimation. The latent trait model is formulate as follows: Let $Y_{ij}$, $j = 1, \ldots, n_i$ be LBW information from region $i = 1, 2, \ldots, 11$

$$Z_{ij} = X_{ij}\beta + v_i + \epsilon_{ij}$$

$$Y_{ij} = g(Z_{ij}),$$

where $\epsilon_{ij} \overset{\text{iid}}{\sim} N(0, \sigma^2)$, $i = 1, \ldots, 11$, $j = 1, \ldots, n_i$ is an $n_i \times 1$ vector of error terms, $\boldsymbol{Y_i} = (y_{i1}, \ldots, y_{in_i})$ is $n_i \times 1$ response variable and $\boldsymbol{z_i} = (z_{i1}, \ldots, z_{in_i})$ $n_i \times 1$ latent variable. Here in this model, $\boldsymbol{X_i}$ is the matrix $n_i \times p$ of independent variables for the $i^{th}$ region of observation, where $i = 1, 2, \ldots, 11$. $\boldsymbol{\beta}$ represents a vector of fixed effects with dimensions $p \times 1$. The latent trait $v_i$ is used to effectively capture geographically unstructured heterogeneity.

Although $Z_i$ cannot be observed directly, the data contain relevant information about it that does not require the specification of $\boldsymbol{g}(.)$. [23] states that when $y_1$ is greater than $y_2$, it follows that $g(Z_1)$ is also greater than $g(Z_2)$ based on the observed data. Considering that $g$ is a non-

decreasing function, this implies that $Z_1$ must be greater than or equal to $Z_2$. Upon observation of $Y = y$, $Z_i$'s must be classified within the designated set:

$$R(y) = \{Z \in \Re^n : z_{i_1} < z_{i_2} \quad \text{if} \quad y_{i_1} < y_{i_2}, \forall i_1, i_2 = 1, \ldots, n_i\}.$$

This demonstrates the feasibility of performing posterior inference based on the understanding that $Z \in R(y)$. In such a circumstance, the posterior density of $\boldsymbol{\beta}$ can be represented as

$$\mathcal{P}(\boldsymbol{\beta} \mid Z \in R(y), \boldsymbol{v}) \propto \mathcal{P}(Z \in R(y), \boldsymbol{v} \mid \boldsymbol{\beta})p(\boldsymbol{\beta}) = p(\boldsymbol{\beta}) \int_{R(y)} \prod_{i=1}^{m} \mathcal{N}_p(X\boldsymbol{\beta} + v_i, \boldsymbol{\Sigma})dz_i$$

In the equation provided above, the term $\mathcal{N}_p(X\boldsymbol{\beta} + v_i, \boldsymbol{\Sigma})$ represents the normal distribution with a mean of $X\boldsymbol{\beta} + v_i$ and a covariance matrix of $\Sigma$. The probability $\mathcal{P}(Z \in R(y), \boldsymbol{v} \mid \boldsymbol{\beta})$, when considered in relation to both $\boldsymbol{\beta}$ and $\boldsymbol{v}$, is called the marginal rank likelihood.

For the variable LBW $Y$, it is possible to obtain information on the fixed effect $\boldsymbol{\beta}$ from $\mathcal{P}(Z \in R(y), \boldsymbol{v} \mid \boldsymbol{\beta})$ without explicitly specifying $g(.)$. This characteristic has notable implications from a computational perspective. When employing a Gibbs algorithm and given a current value, the conditional density $\mathcal{P}(\boldsymbol{\beta} \mid Z \in R(y), \boldsymbol{v})$ can be simplified to $\mathcal{P}(\boldsymbol{\beta} \mid Z = z, \boldsymbol{v})$ as knowing the precise value of $Z$ provides more information than simply knowing that $Z$ is within the range $R(y)$.

Consequently, the conditional distribution regarding $\boldsymbol{\beta}$ depends exclusively on $z$, and $\boldsymbol{v}$, and it adheres to the equation $\mathcal{P}(\boldsymbol{\beta} \mid z, \boldsymbol{v}) \propto \mathcal{P}(z, \boldsymbol{v} \mid \boldsymbol{\beta})\mathcal{P}(\boldsymbol{\beta})$. Given $\boldsymbol{\beta} \mid Z \in R(y), \boldsymbol{v}$, the density of $Z_i$ is proportional to a normal distribution, but is constrained by the condition $Z \in R(y)$. This indicates that $Z_i$ must fall within the specified range. The distribution that generates latent data can be expressed as follows. $Z \mid \boldsymbol{\beta}, v_i \sim \mathcal{N}_{[a_{ij}, b_{ij}]}(X\boldsymbol{\beta} + v_i, \boldsymbol{\Sigma})$. If the values $a_{ij}$ and $b_{ij}$ represent the lower and upper bounds of an interval, then $Z$ is subject to the following constraints:

$$\max\{z_{jh} : y_{jh} < y_{ij}\} < Z_{ij} < \min\{z_{jh} : y_{ij} < y_{jh}\}$$

Various priors, including conjugate and non-informative ones, were taken into account for the model parameters. The regression coefficients ($\boldsymbol{\beta}$) are given a non-informative prior, $\pi(\boldsymbol{\beta}) \propto \mathbf{1}$. It is assumed that the latent traits ($v_i$) follow a normal prior, $v_i \mid \sigma_i^2 \sim \mathcal{N}(0, \sigma_i^2), i = 1, \ldots, 11$, and the scaled inverse chi-square hyperprior is assumed to be used for the variances associated with the latent trait, $\sigma_i^2 \sim \text{Scaled} - \text{Inv} - \chi^2(\omega, \lambda^2)$, where $\omega$ denotes the degree of freedom parameter, while $\lambda^2$ represents the scale parameter. The Markov chain Monte Carlo (MCMC) simulation technique was utilized to generate posterior samples based on the conditional distributions of the model parameters. To evaluate the convergence of the simulated sequences within the models, we meticulously reviewed various plots, including trace plots, density plots, and autocorrelation plots. Furthermore, as part of our analysis, we employ formal convergence tests, such as the potential scale reduction factor (PSRF) test suggested by [24].

**Evaluation metrics.** To evaluate the performance of the models, we examined metrics such as mean absolute error (MAE), mean square error (MSE), and coverage probability at a 95% confidence level. These metrics were established on the basis of the posterior mean of the model parameters. Therefore, a more accurate estimate will result in the lowest MSE and MAE values, as well as a higher percentage of coverage. Calculating the mean square error (MSE) involved taking the average of the squared difference between the estimated value $\hat{\theta}$ and the true value $\theta$. Therefore, the MSE can be given as follows:

$$\text{MSE} = \frac{1}{n}\sum_{j=1}^{n}(\hat{\theta}_j - \theta_j)^2$$

Similarly, the mean absolute error (MAE) was calculated as the average of the absolute deviation of the $\hat{\theta}$ from $\theta$. Accordingly, the MAE is computed as follows:

$$\text{MAE} = \frac{1}{n}\sum_{j=1}^{n}|\hat{\theta}_j - \theta_j|$$

The last, percent coverage, was defined as the average number of times the 95% model credible interval for $\theta$, including the true value. The 95% coverage of the estimated summary of $\hat{\theta}$ that contains the true value is

$$\text{PC} = \frac{1}{n}\sum_{j=1}^{n}I(\theta_j \in [95\% \text{ CI of } \hat{\theta}_j])$$

## Results

In this section, we analyzed the previously introduced LBW data and presented the results of our analysis based on the latent traits model data analysis techniques. As a reminder, this study aims to estimate the parameters and identify the regional distribution and determinants of LBW in Ethiopia for children aged 0-59 months.

### Descriptive results

The prevalence of LBW is presented in Table 1. The corresponding 95% confidence intervals have been determined using variance estimates obtained by linearization of the Taylor series.

Fig 1 shows that children born to mothers from emerging regions (specifically Afar and Somali) have a higher incidence of LBW, while the percentage of LBW among children born in the capital Addis Ababa is comparatively lower than those of other regions. Furthermore, the data presented in Fig 1 show a decrease in the percentage of LBW as the age of the mother increases.

Fig 2 presents a clear correlation between the number of ANC visits and the reduction in the proportion of LBW in children. It also shows a corresponding decrease in the proportion of LBW as the birth order increases.

### Model comparison

A comparison was made between the estimates of LBW obtained from the model that incorporates latent traits and those obtained from the classical model. The `brms` package in R facilitates the estimation of the classical model through the utilization of a Bayesian probit model, yielding LBW estimates for the model parameters. The evaluation criteria used for the comparison included the root mean square error (RMSE), mean absolute error (MAE), and 95% probability coverage.

**Table 1. Prevalence of LBW indicators among children under five years of age in Ethiopia.**

| Indicators | Prevalence | SE | 95% CI |
|---|---|---|---|
| HBW | 16.48 | 0.50 | [15.51, 17.49] |
| NBW | 42.60 | 0.67 | [41.29,43.93] |
| LBW | 14.04 | 0.47 | [13.13,14.98] |
| VLBW | 8.700 | 0.38 | [7.98, 9.48] |
| ELBW | 18.18 | 0.52 | [17.17,19.22] |

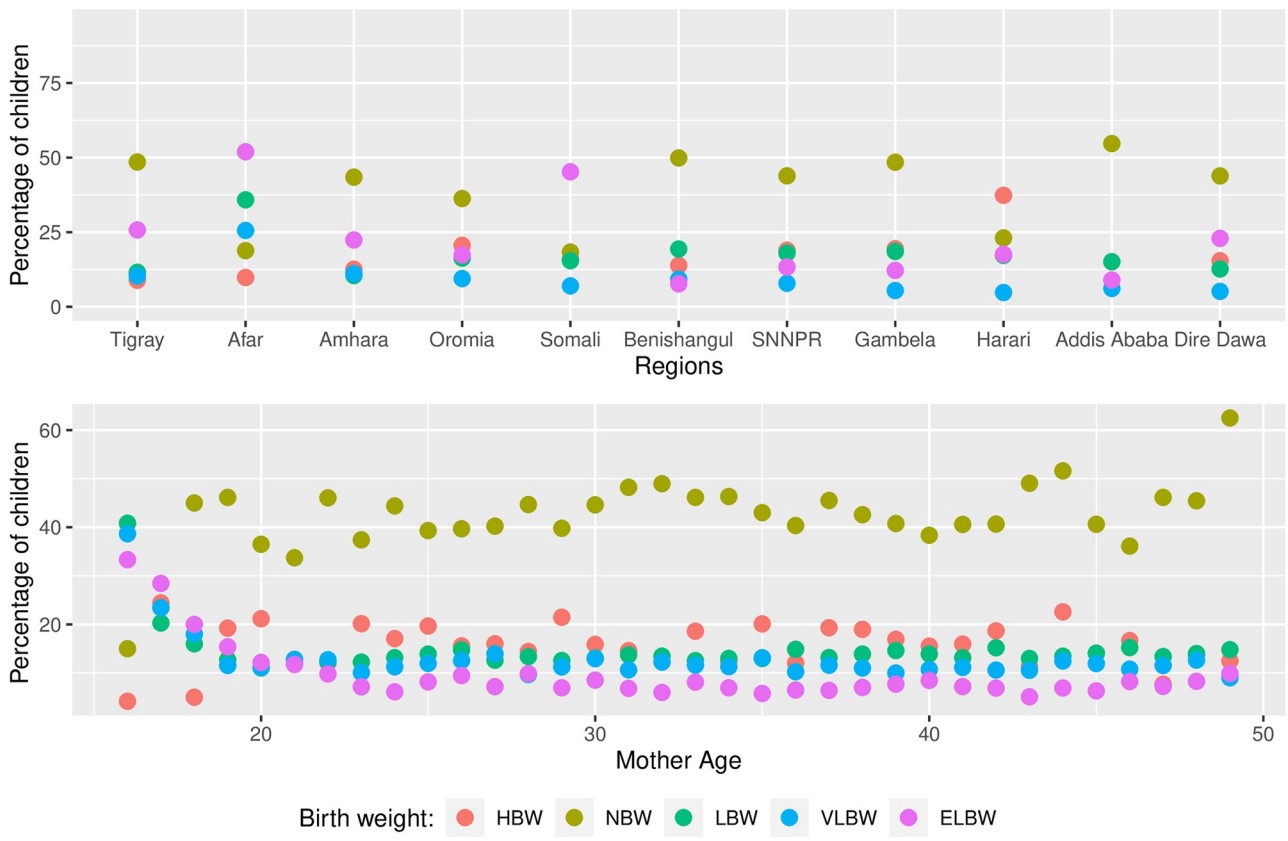

**Fig 1. Percentage of children with LBW by region and mother age.**

The estimates obtained from the developed model demonstrate lower values of RMSE and MAE compared to the estimates of the classical model, indicating the superior performance of the developed model Table 2. Furthermore, the estimate of the developed model showed a 95% higher probability coverage compared to the estimate of the classical model. This finding further supports the notion that the suggested model demonstrates improved performance.

## Analysis of latent traits model

The convergence and independence of the samples were confirmed through various means, including trace graphs (Figs 4 and 5), density graphs (Fig 6), autocorrelation graphs (Figs 7 and 8), and the potential scale reduction factor test (PSRF) for convergence (Table 3). All findings indicate an accurate representation of the stationary distribution with no evidence of assumption violations.

The age of the mother, the number of visits to antenatal care (ANC), the order of birth, and the body mass index (BMI) have a significant influence on low birth weight (LBW), as indicated by the mean posterior beta values of ($\beta_1$= -0.269, $\beta_2$= -0.235, $\beta_3 = -0.120$ and $\beta_5$= -0.257) respectively. This implies that the severity of LBW tends to decrease as the age of the mother, the number of visits to ANC, the order of birth, and the BMI increase. This research study did not find a significant effect of the birth interval on the birth weight of the child.

Fig 3 shows the extent of the variation in LBW between children, both within and between different regions. The figure specifically highlights the highest levels of LBW variation within

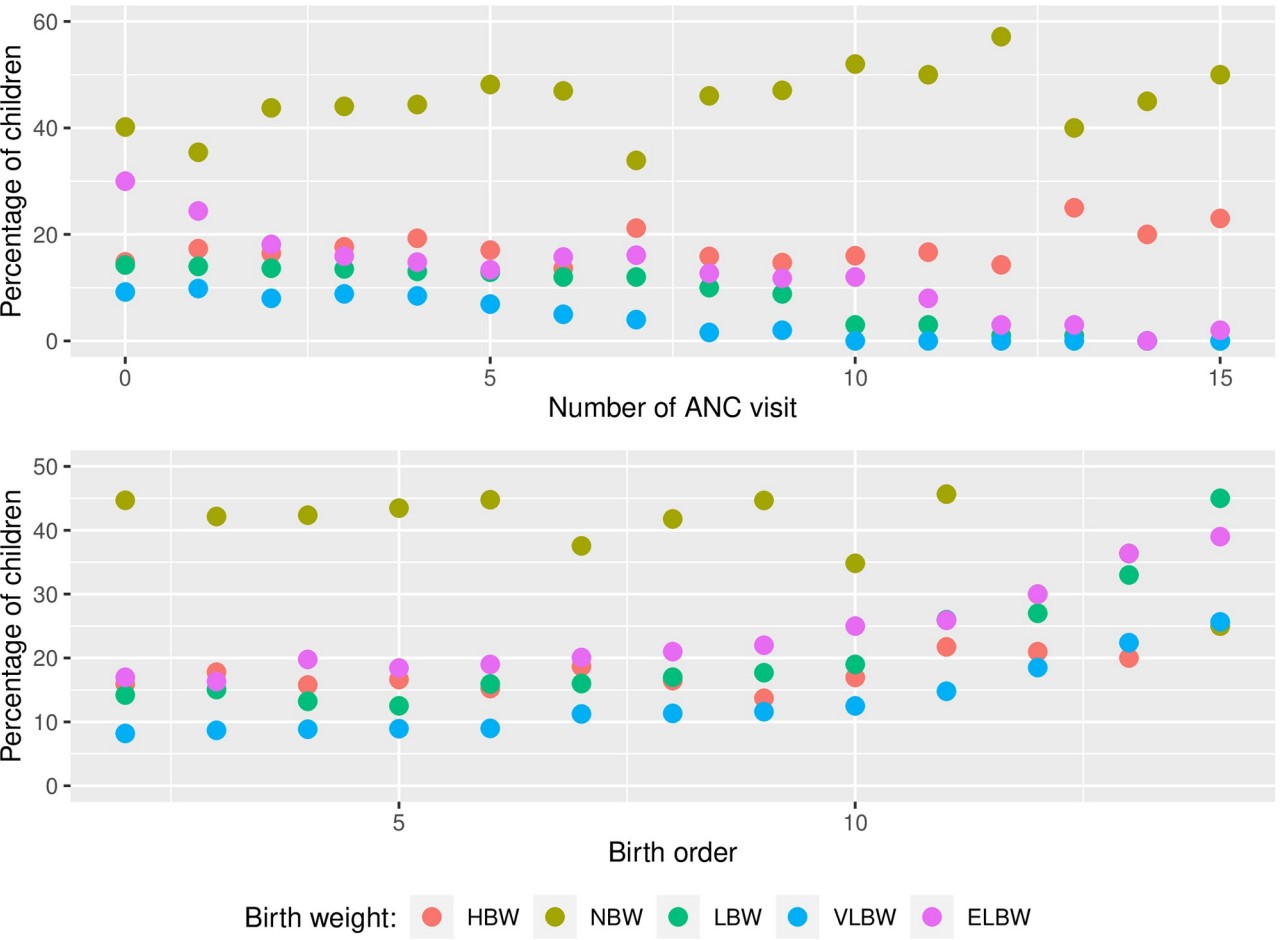

**Fig 2. Percentage of children with LBW by visit to ANC and birth order.**

the region observed in the Afar, Somali and SNNPR regions, as represented by the latent trait parameter of the model. In contrast, the lowest variation in LBW was recorded in the Addis Ababa, Dire Dawa, and Amhara regions.

Figs 4 and 5 show the trace plots of the MCMC samples for the selected model parameters. Each plot shows that the chain effectively samples from its respective stationary distribution, indicating convergence and satisfactory sampling behaviour. The MCMC chain for the fixed effect parameters ($\boldsymbol{\beta}$) and all the latent trait parameters($\boldsymbol{v}$) showed good mixing and rapid convergence.

Similarly, the kernel density plots in Fig 6 indicated that there were no significant convergence issues, as the plots showed an unimodal distribution. Furthermore, (Figs 7 and 8) show the rapid decay of autocorrelation with increasing lag, further supporting the assertion of convergence.

**Table 2. Summary of the RMSE, MAE, and coverage probability values.**

| Models | RMSE | MAE | Coverage probability |
|---|---|---|---|
| Proposed(latent traits model) | 0.123 | 0.137 | 80% |
| Classical model | 0.234 | 0.209 | 60% |

**Table 3. Posterior properties summaries of the latent trait model of LBW.**

| Posterior characteristics | $\beta_1$ | $\beta_2$ | $\beta_3$ | $\beta_4$ | $\beta_5$ | |
|---|---|---|---|---|---|---|
| Posterior mean | -0.269 | -0.235 | -0.120 | 0.169 | -0.257 | |
| Posterior s.e | 0.025 | 0.016 | 0.022 | 0.017 | 0.016 | |
| HPD CI 95% | (-0.320, -0.220) | (-0.268, -0.202) | (-0.162, -0.074) | (-0.032, 0.202) | (-0.291, -0.225) | |
| $\hat{R}$ | 1.000 | 1.000 | 1.000 | 1.000 | 1.000 | |
| Posterior characteristics | $\upsilon_1$ | $\upsilon_2$ | $\upsilon_3$ | $\upsilon_4$ | $\upsilon_5$ | |
| Posterior mean | 4.066 | 4.867 | 4.894 | 4.516 | 4.561 | |
| Posterior s.e | 0.726 | 0.232 | 0.111 | 0.171 | 0.262 | |
| HPD CI 95% | (2.891, 5.144) | (4.493, 5.326) | (4.672, 5.130) | (4.295, 4.938) | (4.222, 5.199) | |
| $\hat{R}$ | 1.004 | 1.002 | 1.003 | 1.003 | 1.002 | |
| Posterior characteristics | $\upsilon_6$ | $\upsilon_7$ | $\upsilon_8$ | $\upsilon_9$ | $\upsilon_{10}$ | $\upsilon_{11}$ |
| Posterior mean | 3.554 | 4.606 | 3.434 | 2.927 | 4.448 | 3.005 |
| Posterior s.e | 0.704 | 0.242 | 0.516 | 0.748 | 1.064 | 1.061 |
| HPD CI 95% | (2.765, 5.217) | (4.064, 4.926) | (2.620, 4.430) | (1.791, 4.382) | (2.271, 5.873) | (0.645, 4.519) |
| $\hat{R}$ | 1.002 | 1.002 | 1.001 | 1.002 | 1.002 | 1.005 |

## Discussion

A latent trait model was proposed to address the parameter estimation problem associated with ordinal response variables. Bayesian rank likelihood estimation was used in the parameter estimation process. In this study, the effectiveness of a suggested latent trait model is compared with the classical model. The performance of both models is evaluated using three metrics: root mean squared error (RMSE), mean absolute error (MAE), and probability coverage (PC). The results showed that the developed model had lower values of RMSE and MAE compared to the classical model. Furthermore, the proposed model had a higher PC than the classical

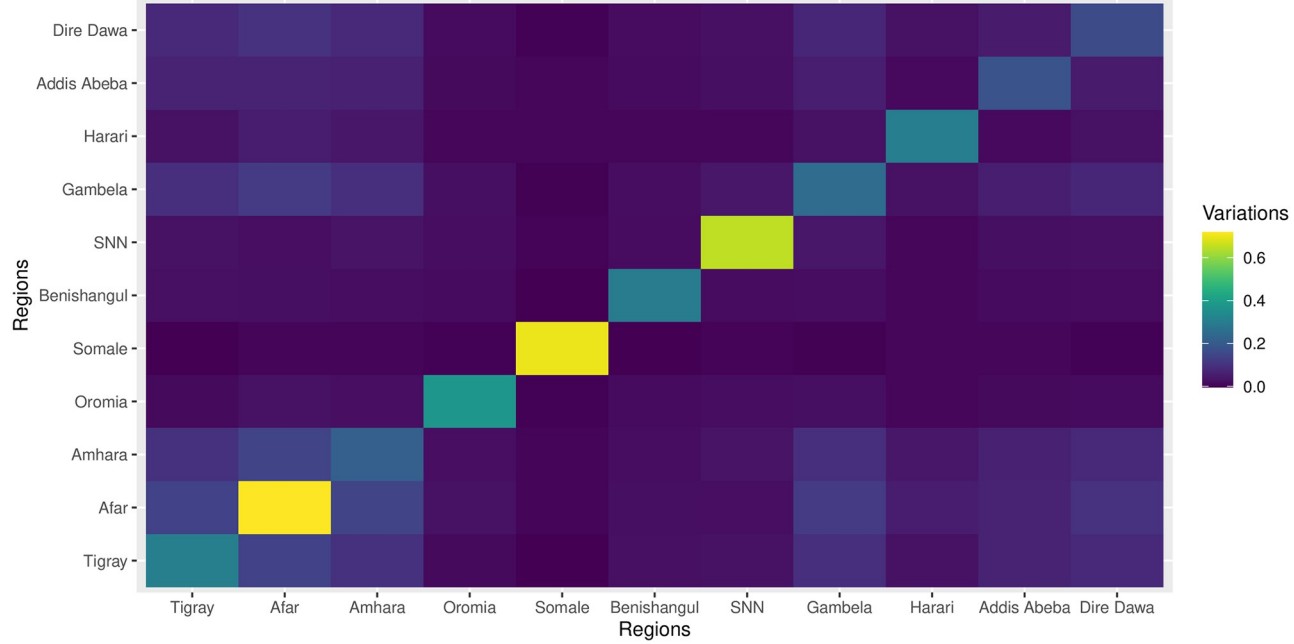

**Fig 3. LBW variations within and between regions based on the latent traits $\upsilon$ parameter.**

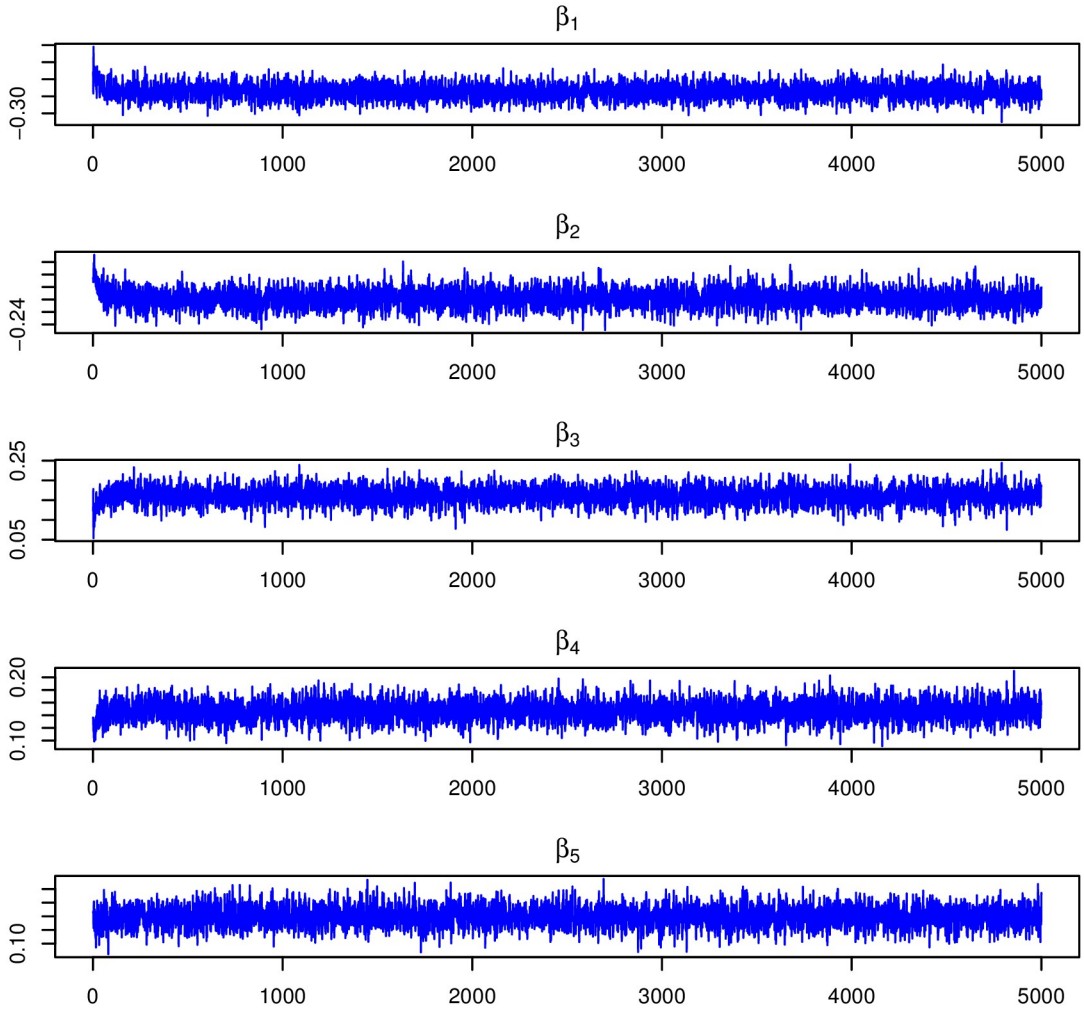

**Fig 4. Trace plots of fixed effects $\beta$ corresponding to the age of the mother, the number of ANC visits, the birth order, the birth interval and the body mass index (BMI).**

one. Based on these results, the latent trait model utilizing Bayesian rank likelihood estimation is considered the better model.

The prevalence of LBW at the national level is reported to be 40.92%, with 14.04% classified as LBW, 8.70% as VLBW and 18.18% as ELBW. These statistics suggest that low birth weight continues to be a major public health issue in the country. These findings are consistent with similar studies conducted in this area [25–27]. There was significant variation within the regions, along with a substantial prevalence of LBW among children under the age of five in Ethiopia. The inclusion of a latent trait effect during the estimation process was determined to be crucial in the study.

Consequently, the prevalence of LBW was estimated using the latent trait model with the Bayesian rank likelihood estimation method. Data visualization graphs and model estimate results indicate that factors such as the age of the mother, antenatal care visits, birth order, and BMI have a significant impact on LBW. Furthermore, there is evidence of regional variation in LBW within the studied regions. There is the highest variation of LBW within the Afar, Somali,

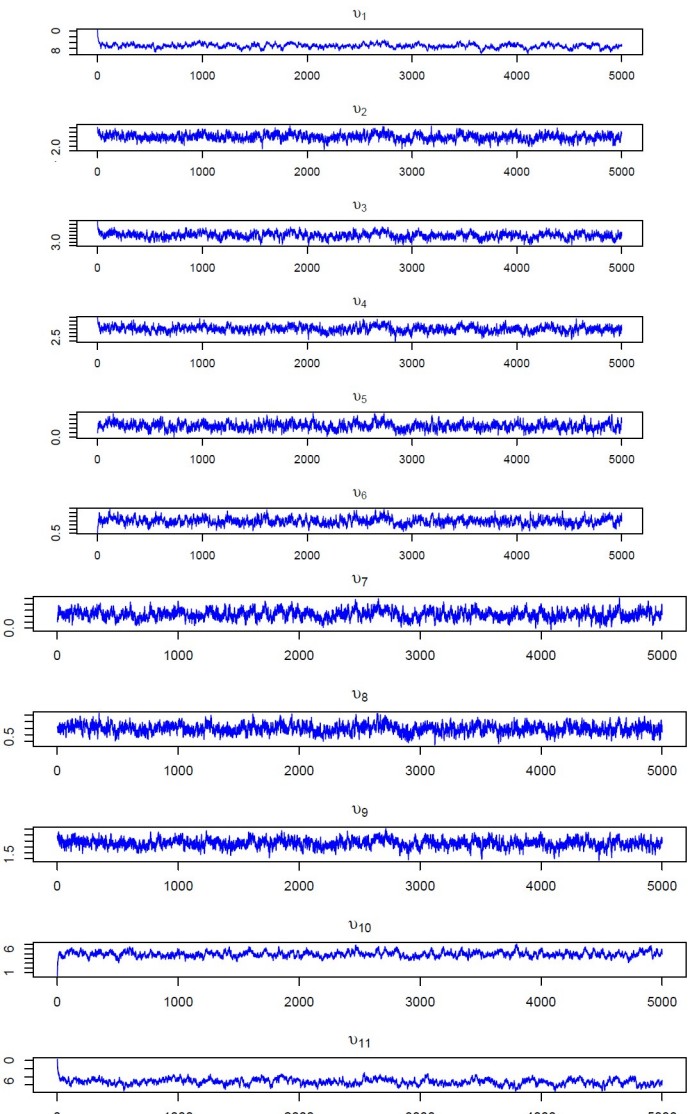

**Fig 5. Trace plots of latent traits $v$ related to Tigray, Afar, Amhara, Oromia, Somali, BenishangulGumuz, SNNPR, Gambela, Harari, Addis Ababa and Dire Dawa, respectively.**

and SNNPR regions, while the Addis Ababa, Dire Dawa, and Amhara regions have the lowest variation. The findings of this study on the relationship between the age of mothers and LBW are consistent with the research carried out by [28]. Tessema et al. [28] demonstrated that as the age of mothers increases, there is a decrease in the severity of LBW. This could be attributed to problems such as child marriage and malnutrition among adolescents in Africa [29].

The findings of this study regarding the impact of birth order on LBW are consistent with the research conducted by [30]. Their study highlights the importance of birth order as a crucial factor that influences LBW. According to [29], it is believed that the age of the mother during childbirth could be the reason for this. The observed relationship between a mother's BMI and birth weight is consistent with the findings reported in [31]. According to [31], this relationship could be attributed to the fact that the weight and height of a mother serve as indicators of her nutritional intake, directly affecting the weight of the child at birth.

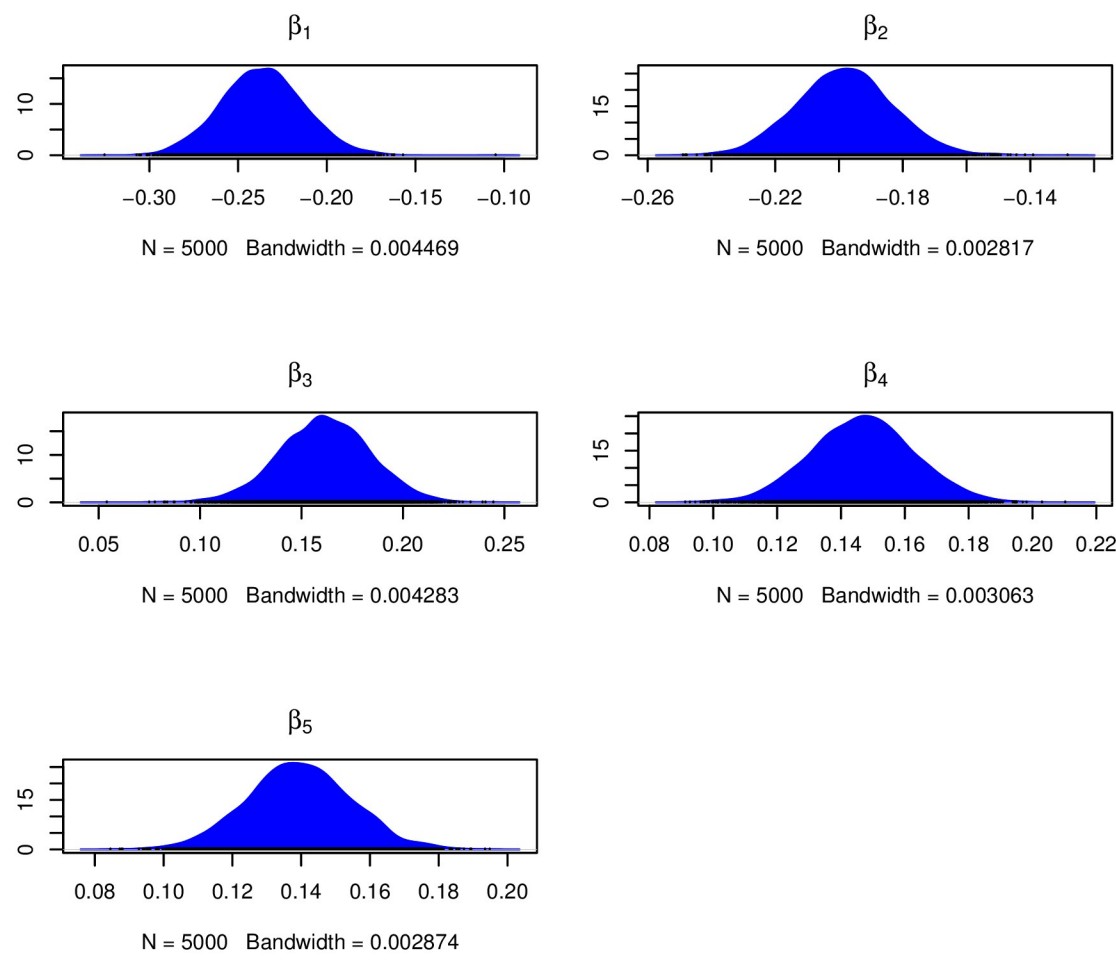

**Fig 6. Kernel density plots of fixed effects β corresponding to the age of the mother, the number of visits to ANC, the order of birth, the birth interval and the body mass index (BMI).**

The results of this study also revealed a significant association between the number of antenatal care visits and LBW. In this study, almost half of the women surveyed, 44.99%, did not attend the ANC. The guidelines of the Federal Ministry of Health on ANC in Ethiopia recommend that women attending ANC receive advice on maintaining a balanced diet and proper nutrition during pregnancy [32–34]. Children whose mothers received a higher number of antenatal care visits were found to have a reduced risk of the severity of LBW. This finding is consistent with the results reported in [35]. This could be because women who have a higher number of ANC visits to healthcare facilities may benefit from increased opportunities for nutritional counseling, as well as iron and folic acid supplementation [1].

Contrary to the results of our study, [28] found an association between LBW and the birth interval. This could be due to sample size differences and their study did not account for possible correlation between groups in their analysis. They also applied standard models to identify predictors of LBW in Ethiopian children.

## Limitations

It is important to consider the limitations of the present study. First, the reliance on self-reported determinants based solely on the mothers' memory introduces the possibility of recall

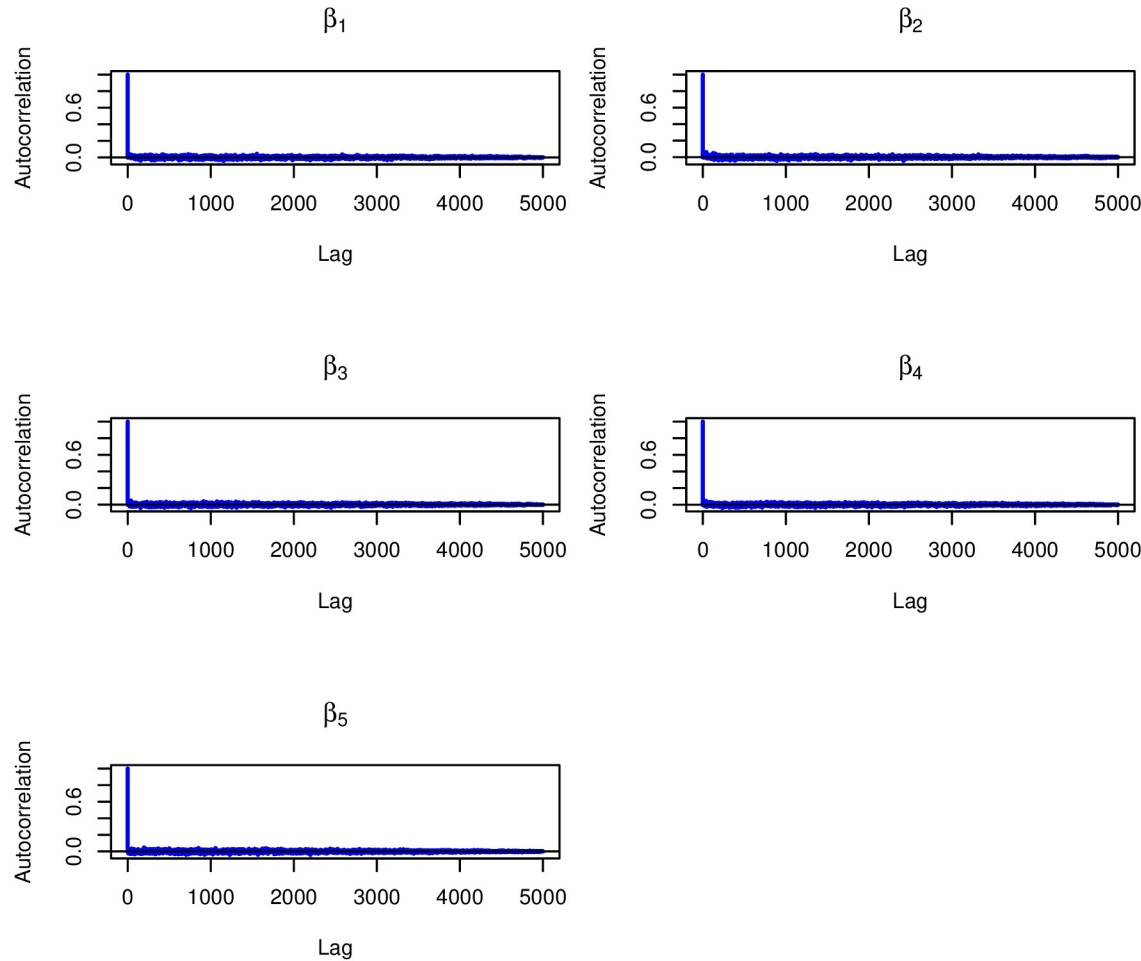

**Fig 7. ACF plots of the fixed effects $\beta$ corresponding to the age of the mother, the number visits to ANC, the order of birth, the birth interval and the body mass index (BMI).**

bias. Furthermore, the use of data from the Ethiopian Demographic and Health Survey (EDHS) that are 7 years old may have influenced the results, as circumstances and conditions could have changed over time. Therefore, the readers are advised to consider these limitations when interpreting the study findings.

## Conclusions

A latent trait model was developed to address the parameter estimation problem associated with ordinal response variables. The Bayesian rank likelihood of estimation was used in the model development process. The efficacy of the suggested model was compared to the existing classical model using RMSE, MAE, and probability coverage (PC). According to the results, the RMSE and MAE of the developed model were smaller than those of the classical model. The PC of the developed model is higher than the classical one. As a result, it can be concluded that the latent trait model utilizing Bayesian rank likelihood estimation is considered the better model. The estimation of LBW among children under five years of age in Ethiopia was carried out at the regional level using the Bayesian rank likelihood estimation method. The incidence of LBW in Ethiopia is significant and shows substantial variation between the different regions

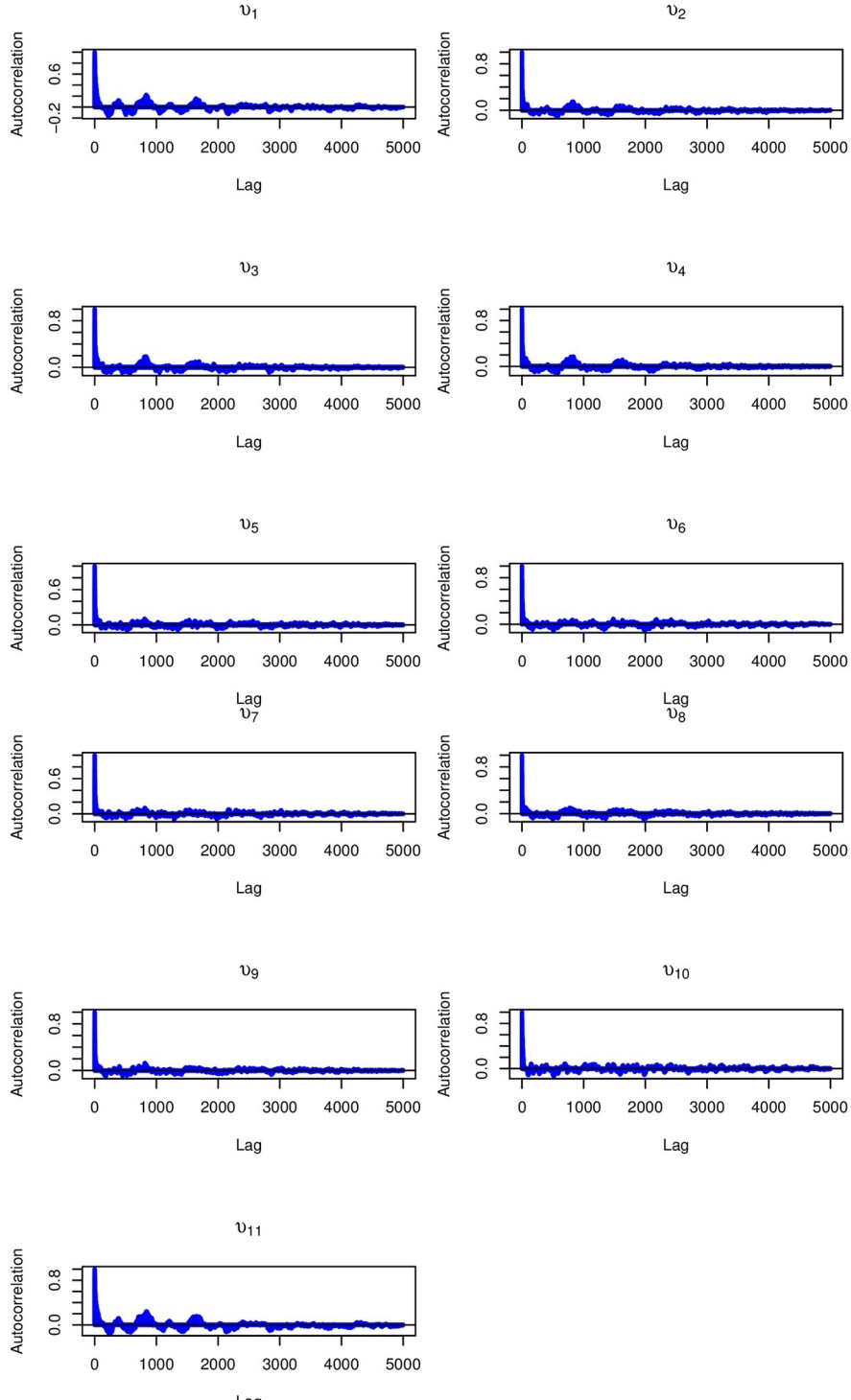

**Fig 8. ACF plots of the latent traits _v_ of Tigray, Afar, Amhara, Oromia, Somali, Benishangul-Gumuz, SNNPR, Gambela, Harari, Addis Ababa, and Dire Dawa, respectively.**

of the country. Therefore, it is crucial to implement targeted public health interventions that reduce LBW among children and increase women's knowledge about LBW in areas where the prevalence of LBW is higher. Thus, ministries, health facilities, and volunteer organizations can use this paper to develop policies to prevent or treat low birth weight in the country.

## Author Contributions

**Conceptualization:** Daniel Biftu Bekalo.

**Data curation:** Daniel Biftu Bekalo, Anthony Kibira Wanjoya, Samuel Musili Mwalili.

**Formal analysis:** Daniel Biftu Bekalo.

**Software:** Daniel Biftu Bekalo, Samuel Musili Mwalili.

**Supervision:** Anthony Kibira Wanjoya, Samuel Musili Mwalili.

**Writing – original draft:** Daniel Biftu Bekalo.

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
