## [Decision Letter · Decision Letter 0]

6 Mar 2024

PONE-D-23-37632Bayesian Rank Likelihood Based Estimation: An Application to Low Birth Weight in EthiopiaPLOS ONE

Dear Dr. Biftu,

Thank you for submitting your manuscript to PLOS ONE. After careful consideration, we feel that it has merit but does not fully meet PLOS ONE’s publication criteria as it currently stands. Therefore, we invite you to submit a revised version of the manuscript that addresses the points raised during the review process.

We look forward to receiving your revised manuscript.

Kind regards,

Abay W. Tadesse

Academic Editor

PLOS ONE

4. We note that your Data Availability Statement is currently as follows: [All relevant data are within the manuscript and its Supporting Information files]

5. Please remove your figures from within your manuscript file, leaving only the individual TIFF/EPS image files, uploaded separately. These will be automatically included in the reviewers’ PDF.

6. Please update your submission to use the PLOS LaTeX template. The template and more information on our requirements for LaTeX submissions can be found at http://journals.plos.org/plosone/s/latex.

Additional Editor Comments:

Dear Dr Biftu,

Thank you for submitting this interesting paper, but there are many issues needs attention before publication.

If you consider the Bayesian model as a novel (applied since 1976), as raised by Reviewer #2, you have to do methodological comparison with conventional models.

I could see that labels for your convergence algorism assessment are missed and confusing for readers. Besides, few your plots are hard to visualize if converged or not. Please provide the details as an appendex.

Additionally, it is not clear which form of Bayesian you applied in your analysis (informative, non-informative).

Major concern, your discussion is too shallow and the possible explanations are not plausible. Please also include implication your study for policy or other context.

Please address the comments raised by the reviewers.

Reviewers' comments:

Reviewer's Responses to Questions

**Comments to the Author**

1. Is the manuscript technically sound, and do the data support the conclusions?

Reviewer #1: Partly

Reviewer #2: Partly

2. Has the statistical analysis been performed appropriately and rigorously? 

Reviewer #1: Yes

Reviewer #2: I Don't Know

3. Have the authors made all data underlying the findings in their manuscript fully available?

Reviewer #1: Yes

Reviewer #2: No

4. Is the manuscript presented in an intelligible fashion and written in standard English?

Reviewer #1: No

Reviewer #2: No

5. Review Comments to the Author

Reviewer #1: Manuscript Number: PONE-D-23-37632

Article Type: Research Article

Full Title: Bayesian Rank Likelihood Based Estimation: An Application to Low Birth Weight in Ethiopia

General

• Overall, a thorough review of the manuscript for grammatical issues is still required. Even within the abstract and introduction, there are multiple grammatical issues that make it difficult to read. Editing by a colleague or writing service with strong English skills is needed.

Abstract:

• It would be better if the abstract could be structured as follows: background, methods, results, and conclusion.

• As a justification for the study, you stated, "Most studies conducted in Ethiopia have utilized standard models employing maximum likelihood estimation with small sample sizes." But it could be better if you describe the limitations of those standard models and the advantage of Bayesian rank likelihood-based estimation over the standard models.

• The abstract should explain at least the data source, the study population, and the sample size.

• The abstract should be a stand-alone document. Some of your statements in the abstract are not standalone. For example, you stated, “The prevalence of low birth weight varies across different regions of the country, highlighting the need for targeted interventions in areas with a higher prevalence." This may need further clarification where the prevalence is high.

• The study revealed significant associations between birth weight and the mother's age, number of antenatal care visits, birth order, and BMI. What is new? The association between low birth weight and those factors is well known and documented by several previous studies. It is better if you magnify or focus on a new and untouched variable.

• As a recommendation, you said, “It is essential to implement community-based health promotion prioritizing antenatal care follow-up and maternal nutrition.” How? Where is the evidence-based intervention (i.e., community-based health promotion intervention)? Why not facility-based?

• Overall, in the abstract, I couldn’t find the answer to your study objective: to estimate parameters and provide a nationwide estimation of low birth weight and its risk factors in Ethiopia. At least you have to present the magnitude of LBW according to your estimation.

Background

• Please provide a comprehensive review of the epidemiology of LBW (magnitude, distribution, and determinants) in Ethiopia.

• In the background, you wrote, “According to [15], in the presence of ordinal variables, a maximum likelihood-based method yields estimates with substantial bias and inaccurate coverage probabilities.” Explain why these models are biased.

• In the background, a bit more detail is needed about the Bayesian rank likelihood method in the latent traits model.

Methods and Materials

• There are nine regions. Are you sure? Currently, there are twelve regional states and two chartered cities (Addis Ababa and Dire Dawa).

• Explain how you extract those variables from EDHS data sets.

• Explain how you managed missing data, if any.

• Describe the purpose of using trace plots, kernel density plots, and ACF plots.

• Have you considered a design-based analysis since the DHS survey employed a complex design?

Results

• When you present findings, start with a text description and refer the reader to any figure or table for details.

• In the result section, present only what you found, not how you found it.

• First, you need to present the characteristics of your study participants.

• Present the overall estimation of LBW before presenting the regional variation.

• Why did you prefer to use figures instead of tables? You may better organize and present several findings shown in Fig. 1, Fig. 2, etc.

• One of your findings is that the severity of LBW tends to decrease as the age of the mother increases. But is that always true among those mothers with an advanced age, which may be greater than 35 years? Have you conducted any sensitivity analyses?

• Overall, the result lacks a proper and clear description of statistical outputs in non-statistical explanations.

Discussion

• Apart from a simple comparison of findings, discussion should entail the clinical and practical implications of the findings.

• As part of the limitations of the study, you need to disclose the role of chance, bias, and confounding factors in your findings. And how do you minimize or control their effects?

• Recommendations are not specific, practical, or action-oriented. (i.e., targeted public health interventions, community-based health promotion activities, etc.)

Reviewer #2: Title: Bayesian Rank Likelihood-Based Estimation: An Application to Low Birth Weight in Ethiopia is a good introduction to Bayesian-based estimation. But I do have some points to be more explained

- Intrauterine is the main factor that causes LBW in developing countries in literature, can you explain this about your findings or your context and provide enough justification for why would it differ in the Ethiopian context? This is important because the connection with literature is vital.

- Additionally, using Bayesian estimation by itself is not enough gap to conduct research, the researchers must explore more to declare a clear gap as enough evidence on LBW is already available in Ethiopia.

- Your introduction section did not reflect the fact that many studies were conducted from different perspectives, you may need to dig more to show the gap in the literature.

- If you consider the Bayesian model as a novel (I reviewed many papers with Bayesian models), you need to present more methodological differences and provide enough evidence of how it is different from other traditional models to lay ground for readers

- Variables in finding can be rearranged to a positive or negative association with LBW

- Until in the method you did not provide evidence of source of data that is a big missing, please correct accordingly starting from abstract

- The data you used is indeed from DHS 2016, isn’t this too old? There is a recent one in 2019. Or why you can not combine them to see the commutative effect or just use the recent Mini-DHS.

- Methods are very limited as you said a golden approach, may need to include all method and material sections rather than present context and directly go to statistical approaches e.g. data preparation, sampling, and handling missings….generally, It is good to have a similar structure between the results and the method section.

- What is the difference between Birth weights such as HBW NBW LBW VLBW ELBW and what does progress look like? (maybe in the table) because in Figure 1, I can see no difference among LBW, VLBW, and ELBW

- You need improve the discussion as per the above comments

- Conclusion must be more practical than tending to provide more general information.

-

6. PLOS authors have the option to publish the peer review history of their article (what does this mean?). If published, this will include your full peer review and any attached files.

Reviewer #1: **Yes: **Yimer Seid Yimer, Assistant Professor, School of Public Health, Addis Ababa University, Ethiopia.

Reviewer #2: **Yes: **Girma Gilano

---

## [Author Response · Author response to Decision Letter 0]

22 Mar 2024

Responses to the academic editor and reviewers' comments

First of all, we would like to say thank you for your constructive and very important comments. We accepted the comments and made appropriate changes. 

Response to the academic editor 

• If you consider the Bayesian model as a novel (applied since 1976), as raised by Reviewer #2, you have to do a methodological comparison with conventional models.

We conducted a methodological comparison based on the feedback provided by the reviewer and academic editor.

• I could see that labels for your convergence algorism assessment are missing and confusing for readers. Besides, a few of your plots are hard to visualize if converged or not. 

We explained what those plots indicate in the paper's results section.

• Additionally, it is not clear which form of Bayesian you applied in your analysis (informative, non-informative). 

We used both informative and non-informative priors for the model parameters in our paper on page 5.

• Major concern, your discussion is too shallow and the possible explanations are not plausible. Please also include implications of your study for policy or other context.

We took the comments seriously and made an effort to incorporate them as best we could.

Response to Reviewer 1

• Overall, a thorough review of the manuscript for grammatical issues is still required. Even within the abstract and introduction, multiple grammatical issues make it difficult to read. Editing by a colleague or writing service with strong English skills is needed.

We have sent the paper to a colleague for language editing.

• It would be better if the abstract could be structured as follows: background, methods, results, and conclusion.

We wrote the abstract according to the comment 

• As a justification for the study, you stated, "Most studies conducted in Ethiopia have utilized standard models employing maximum likelihood estimation with small sample sizes." However, it could be better if you describe the limitations of those standard models and the advantage of Bayesian rank likelihood-based estimation over the standard models.

We described the limitations of those standard models in the abstract as per the comment 

• The abstract should explain at least the data source, the study population, and the sample size.

The data source, study population and sample size are explained in the abstract accordingly 

• The abstract should be a stand-alone document. Some of your statements in the abstract are not standalone. For example, you stated, “The prevalence of low birth weight varies across different regions of the country, highlighting the need for targeted interventions in areas with a higher prevalence." This may need further clarification where the prevalence is high.

We tried to address the comment as per the comment 

• Overall, in the abstract, I couldn’t find the answer to your study objective: to estimate parameters and provide a nationwide estimation of low birth weight and its risk factors in Ethiopia. At least you have to present the magnitude of LBW according to your estimation.

We tried to address the comment as per the comment 

• The study revealed significant associations between birth weight and the mother's age, number of antenatal care visits, birth order, and BMI. What is new? The association between low birth weight and those factors is well-known and documented by several previous studies. It is better if you magnify or focus on a new and untouched variable. 

After reviewing the available literature, we included potential risk factors for LBW and employed a new methodology of parameter estimation.

• Explain how you managed missing data, if any.

We only analysed complete cases in our analysis.

• Describe the purpose of using trace plots, kernel density plots, and ACF plots.

We explained what these plots indicate based on the output of results in the paper as per the reviewer's comment 

• Have you considered a design-based analysis since the DHS survey employed a complex design?

We applied the latent trait model that accounts for within and between-group variation while considering the multi-stage sampling design of the data.

• Present the overall estimation of LBW before presenting the regional variation.

We present the overall estimations in Tables 1 and 2 before presenting the regional estimation.

• In the result section, present only what you found, not how you found it.

We removed and corrected it accordingly 

• Why did you prefer to use figures instead of tables? You may better organise and present several findings shown in Fig. 1, Fig. 2, etc

Since most covariates used in the study are continuous variables, scatter plots are better than tables to show the pattern.

• Overall, the result lacks a proper and clear description of statistical outputs in non-statistical explanations.

We have tried to correct this by considering the comment of the reviewer 

• One of your findings is that the severity of LBW tends to decrease as the age of the mother increases. But is that always true among those mothers with an advanced age, which may be greater than 35 years? Have you conducted any sensitivity analyses?

We analysed the data and found that LBW severity decreases with age our finding also agrees with the findings of other authors 

• Overall, the result lacks a proper and clear description of statistical outputs in non-statistical explanations.

We have made an effort to include all the comments accordingly.

• Apart from a simple comparison of findings, discussion should entail the clinical and practical implications of the findings.

We tried to incorporate the clinical and practical implications of the findings as per the reviewer's comment 

• Recommendations are not specific, practical, or action-oriented. 

We modified our recommendation based on our findings

Response to Reviewer 2

• Intrauterine is the main factor that causes LBW in developing countries in literature, can you explain this about your findings or your context and provide enough justification for why would it differ in the Ethiopian context? This is important because the connection with literature is vital. 

Since we could not get enough evidence/ literature that indicated Intrauterine as the main factor of LBW we just removed it from the background.

• Additionally, using Bayesian estimation by itself is not enough gap to conduct research, the researchers must explore more to declare a clear gap as enough evidence on LBW is already available in Ethiopia. 

We utilized the Bayesian method for parameter estimation as LBW is an ordinal qualitative variable. The MLE method can lead to biased and inaccurate probability coverage due to the presence of latent variables, hence, the Bayesian method was preferred.

• Your introduction section did not reflect the fact that many studies were conducted from different perspectives, you may need to dig more to show the gap in the literature. 

We have tried to add studies conducted on rank likelihood by different authors as per the comment 

• If you consider the Bayesian model as a novel (I reviewed many papers with Bayesian models), you need to present more methodological differences and provide enough evidence of how it is different from other traditional models to lay ground for readers. 

We tried to compare the proposed model and the existing model (classical one) and showed the difference as per the comment of the reviewer 

• Until in the method you did not provide evidence of source of data that is a big missing, please correct accordingly starting from abstract

We incorporated the source of our data as per the comment 

• The data you used is indeed from DHS 2016, isn’t this too old? There is a recent one in 2019. Or why you can not combine them to see the commutative effect or just use the recent Mini-DHS.

We used the EDHS 2016 because LBW data is not available in Mini-EDHS 2019.

• Methods are very limited as you said a golden approach, may need to include all method and material sections rather than present context and directly go to statistical approaches e.g. data preparation, sampling, and handling missing…. generally, It is good to have a similar structure between the results and the method section.

We incorporated the comments accordingly 

• What is the difference between Birth weights such as HBW NBW LBW VLBW ELBW and what does progress look like? (maybe in the table) because in Figure 1, I can see no difference among LBW, VLBW, and ELBW

In Table 1, we have tried to show the distribution of all levels of LBW and in Figure 1 we tried to show the pattern of LBW by region and age of mothers 

• You need to improve the discussion as per the above comments

We improved the discussion as per the comments 

• Conclusion must be more practical than tending to provide more general information.

We tried to make our conclusion more practical as per the comment

---

## [Decision Letter · Decision Letter 1]

25 Apr 2024

PONE-D-23-37632R1Bayesian Rank Likelihood-Based Estimation: An Application to Low Birth Weight in EthiopiaPLOS ONE

Dear Dr. Bekalo,

Thank you for submitting your manuscript to PLOS ONE. After careful consideration, we feel that it has merit but does not fully meet PLOS ONE’s publication criteria as it currently stands. Therefore, we invite you to submit a revised version of the manuscript that addresses the points raised during the review process.

We look forward to receiving your revised manuscript.

Kind regards,

Abay W. Tadesse

Academic Editor

PLOS ONE

Additional Editor Comments:

Dear authors,

The reviewers have raised significant and critical issues concerning this manuscript and therefore the manuscript is not acceptable for publication in its current form. If a revised manuscript is submitted for consideration for publication the Reviewers’ concerns must be fully addressed.

Reviewers' comments:

Reviewer's Responses to Questions

**Comments to the Author**

1. If the authors have adequately addressed your comments raised in a previous round of review and you feel that this manuscript is now acceptable for publication, you may indicate that here to bypass the “Comments to the Author” section, enter your conflict of interest statement in the “Confidential to Editor” section, and submit your "Accept" recommendation.

Reviewer #1: All comments have been addressed

Reviewer #2: (No Response)

2. Is the manuscript technically sound, and do the data support the conclusions?

Reviewer #1: Yes

Reviewer #2: Yes

3. Has the statistical analysis been performed appropriately and rigorously? 

Reviewer #1: Yes

Reviewer #2: Yes

4. Have the authors made all data underlying the findings in their manuscript fully available?

Reviewer #1: Yes

Reviewer #2: No

5. Is the manuscript presented in an intelligible fashion and written in standard English?

Reviewer #1: Yes

Reviewer #2: Yes

6. Review Comments to the Author

**Reviewer #1:** (No Response)

**Reviewer #2:** TITLE: Bayesian Rank Likelihood Based Estimation: An Application to Low Birth Weight in Ethiopia

Very good title, but it still has many concerns that need serious attention. I am sorry for the delay but this manuscript may need some critical evaluation and is far from consideration for publication. My comments are presented as follows

- The authors already published an article with similar findings from previous EDHS data(https://doi.org/10.5061/dryad.3j9kd51sg). This makes the reasoning for conducting the current study insufficient. What is the difference and how current recommendation show improvement from the previous one

- In paragraphs 3 and 4 all the determinants factors were presented very well. Thus, if all is known about LBW, what is the importance of having the current study?

- The argument of a large sample covering a larger area of the country, even the study the authors published in March 2024 is larger and covered a more area of the country. so, how did this work out for the authors without making a clear link between this and the currently published one?

- Overall, the evidence on LBW is overly available already, and the current argument that because of using Bayesian Rank Likelihood Based Estimation needs to be carefully evaluated if it deserves research. The authors need to sequentially construct the flow of information including the most current evidence and show why it is important to research in the background/introduction.

- For me, I can't make any difference between the published article (https://doi.org/10.5061/dryad.3j9kd51sg) and the currently under review manuscript.

- It looks to me completely similar to what was published in March 2024 (same dependent variable, the same database, the same population but different articles)….how that is possible. I would retake the revision in case I get somehow wrong.

7. PLOS authors have the option to publish the peer review history of their article (what does this mean?). If published, this will include your full peer review and any attached files.

Reviewer #1: **Yes: **Yimer Seid Yimer

Reviewer #2: No

---

## [Author Response · Author response to Decision Letter 1]

26 Apr 2024

Responses to the reviewers' comments

First of all, we would like to say thank you for your comments. 

Response to Reviewer 2

• The authors already published an article with similar findings from previous EDHS data(https://doi.org/10.5061/dryad.3j9kd51sg). This makes the reasoning for conducting the current study insufficient. What is the difference and how does the current recommendation show improvement from the previous one?

We would like to express our gratitude to the reviewer for reviewing our paper. However, we would like to clarify that we have not previously published any similar findings to the current study. We have made the data we used for our current study publicly available on the DRYAD repository, as per the guidelines of PLOS ONE. The data is accessible to everyone and can be used to gain further clarification about our study. We deposited a dataset to a public repository (DRYAD) and included the study's abstract and data description. The data repository team requested us to provide a title for the data that we deposited in DRYAD. We gave it a title that is almost identical to the title of the study and they published it online on the 28th of March 2024. 

• The argument of a large sample covering a larger area of the country, even the study the authors published in March 2024 is larger and covered a more area of the country. so, how did this work out for the authors without making a clear link between this and the currently published one? 

It is the data that we have used for the current study that was published in March 2024 not the full study. This is the link to the published dataset Rank likelihood-based estimation of low birth weight in Ethiopia (zenodo.org)

• Overall, the evidence on LBW is overly available already, and the current argument that because of using Bayesian Rank Likelihood Based Estimation needs to be carefully evaluated if it deserves research. The authors need to sequentially construct the flow of information including the most current evidence and show why it is important to research in the background/introduction.

In the background/introduction, we explained the importance of using the Bayesian rank likelihood method instead of the likelihood method. When dealing with ordered categorical variables, using the likelihood method can result in a biased estimator and inaccurate coverage probability. To address this issue, we introduced the Bayesian method of parameter estimation for our current study. 

• - For me, I can't make any difference between the published article (https://doi.org/10.5061/dryad.3j9kd51sg) and the currently under review manuscript.


https://doi.org/10.5061/dryad.3j9kd51sg) is the published data taken from the current study, it is not an article. 

• It looks to me completely similar to what was published in March 2024 (same dependent variable, the same database, the same population but different articles). how that is possible. I would retake the revision in case I get somehow wrong.

We understand the confusion caused by the publication of the data before the study itself. 

N.B: We deposited the dataset we used in the study in the public repository DRYAD as per the guideline of the PLOS ONE

---

## [Editor Report · Decision Letter 2]

30 Apr 2024

Bayesian Rank Likelihood-Based Estimation: An Application to Low Birth Weight in Ethiopia

PONE-D-23-37632R2

Dear Dr. Bekalo,

We’re pleased to inform you that your manuscript has been judged scientifically suitable for publication and will be formally accepted for publication once it meets all outstanding technical requirements.

Kind regards,

Abay W. Tadesse

Academic Editor

PLOS ONE
---

## [Editor Report · Acceptance letter]

10 May 2024

PONE-D-23-37632R2 

PLOS ONE

Dear Dr. Bekalo, 

I'm pleased to inform you that your manuscript has been deemed suitable for publication in PLOS ONE. Congratulations! Your manuscript is now being handed over to our production team.

Kind regards, 

on behalf of

Mr. Abay Woday Tadesse 

Academic Editor

PLOS ONE